# A Variational Framework for Graph Generation with Fine-Grained Topological Control

## Abstract

Controlled graph generation is the process of generating graphs that satisfy specific topological properties (or attributes). Fine-grained control over graph properties allows for customizing generated graphs to precise specifications, which is essential for understanding and modeling complex networks. Existing approaches can only satisfy a few topological properties such as number of nodes or edges in output graphs. This paper introduces CGRAPHGEN, a novel conditional variational autoencoder that, unlike existing approaches, uses graph adjacency matrix *during* training, along with the desired graph properties, for improved decoder tuning and precise graph generation, while relying only on attributes during inference. In addition, CGRAPHGEN implements an effective scheduling technique to integrate representations from both adjacency matrix and attribute distributions for precise control. Experiments on five real-world datasets show the efficacy of CGRAPHGEN compared to baselines, which we attribute to its use of adjacency matrix during training and effective integration of representations, which aligns graphs and their attributes in the latent space effectively and results in better control.

## 1 Introduction

Graph generation is the process of generating graphs that mimic real world structures, e.g. following a power-law distribution in terms of node degree (Erdös & Rényi, 1959; Barabási & Albert, 1999; You et al., 2018). Controlled graph generation is the process of generating graphs that satisfy specific topological attributes (Zahirnia et al., 2024; Martinkus et al., 2022), with applications in drug discovery (e.g. generating new molecules that satisfy certain chemical properties) (Jin et al., 2018; Shi et al., 2019; Jin et al., 2020; Luo et al., 2021; Popova et al., 2019; Shi et al., 2020; Liu et al., 2021; Zang & Wang, 2020; De Cao & Kipf, 2018), synthetic material design (Wang et al., 2022; Sanchez-Lengeling & Aspuru-Guzik, 2018), simulating social network and social interactions (Pitas, 2016; Zhou et al., 2020; Zeno et al., 2021), program graph generation from source codes (Allamanis et al., 2018), and completing knowledge graphs (Melnyk et al., 2022; Zhou et al., 2023; Cao et al., 2023).

Despite significant progress in graph generation, existing works often lack fine-grained control over structural attributes. Most approaches focus on a limited set of graph attributes as controls (Zahirnia et al., 2024; Chen et al., 2023; Martinkus et al., 2022). In particular, Zahirnia et al. (2024) proposed a variational autoencoder that learns a latent adjacency matrix from statistics like number of edges, triangles, random walks and $k$-hop neighbors. Chen et al. (2023) proposed a discrete diffusion model by explicitly focusing on node degrees to control graph generation. Similarly, Madeira et al. (2024) built on discrete diffusion techniques to incorporate specific graph properties such as planarity or acyclicity to generate graphs. Finally, Martinkus et al. (2022) proposed a model based on generative adversarial networks to control graph generation by focusing on eigenvalues and eigenvectors as more abstract topological properties.

In this paper, we propose **C**ontrolled **Graph Gen**eration (CGRAPHGEN), a novel end-to-end conditional variational autoencoder for generating graphs that satisfy fine-grained topological attributes. CGRAPHGEN implements an effective scheduling technique that integrates representations from both adjacency matrix and attribute distributions to enable more fine-grained and precise control for graph generation.

The closet approach to ours is GenStat (Zahirnia et al., 2024), which is a standard autoencoder model for controlled graph generation. It encodes given graph attributes to produce a latent adjacency matrix, which is then used by a decoder to produce the attributes. CGRAPHGEN differs from GenStat from several aspects: it uses both graph attributes and the adjacency matrix of graphs *during* training for improved decoder tuning, while relying only on attributes during inference; it introduces a novel scheduling technique to integrate latent representations from adjacency matrix and attribute distributions for effective training; and, unlike previous approaches, it can handle any number of fine-grained control attributes without any modification, which provides flexibility in graph generation.

The contributions of the paper are:

- CGRAPHGEN, a novel conditional variational autoencoder to generate graphs conditioned on topological attributes. It uses both graph adjacency matrix and attribute vectors during training for precise graph generation, while relying only on attributes during inference.
- MIXTURE-SCHEDULER, a novel scheduling technique to effectively integrate representations from adjacency matrix and attribute distributions for effective fine-grained topological control in generation.

We evaluate CGRAPHGEN across multiple datasets and our results show its efficacy across the datasets compared to baselines. We find that mixing attribute representations from prior distribution and adjacency matrix representations from posterior distributions helps align graphs and their attributes in the latent space effectively and results in better control. In addition, slower rates of including information from the prior helps more accurate graph generation. Lastly, we find that increasing the number of control attributes helps the model with more precise graph generation. [1]

## 2 CONTROLLED GRAPH GENERATION

**Problem Definition** Given a graph attribute vector $\mathbf{c}$, which represents fine-grained information about the topology of a target graph $G$ (see examples of these attributes in §2.1), we aim to generate a graph $\hat{G}$ such that its structure satisfies the desired attributes in $\mathbf{c}$.

**Solution Overview** During training, CGRAPHGEN uses the adjacency matrix $\mathbf{A}$ of the target graph $G = (V, E)$ and its corresponding attribute vector $\mathbf{c}$ to learn controlled graph generation, i.e. joint distributions of training graphs and their attributes. At the inference time, however, CGRAPHGEN only uses control attribute vector to generate graphs that satisfy the given attribute vectors. As Figure 1 shows, CGRAPHGEN encodes the structural representation $\mathbf{Z_G}$ from adjacency matrix $\mathbf{A}$ to parameterize the posterior distribution $q_\phi$, and the attribute representation $\mathbf{Z_c}$ from the attribute vector $\mathbf{c}$ to build prior distribution $p_\theta$. CGRAPHGEN samples from distributions $q_\phi$ and $p_\theta$ to combines $\mathbf{Z_G}$ and $\mathbf{Z_c}$ using MIXTURE-SCHEDULER to obtain the latent representation $\mathbf{Z}$, which balances structural and attribute information. The MIXTURE-SCHEDULER aims to bring posterior $q_\phi$ and $p_\theta$ closer to each other, as they represent graphs with the same topological structure. The decoder then learns the likelihood distribution $p_\psi$ from $\mathbf{Z}$ to generate a graph $\hat{G}$ that satisfies the attribute vector $\mathbf{c}$. At inference time, CGRAPHGEN relies only on the prior distribution $p_\theta$ and likelihood $p_\psi$ to condition graph generation based on attribute vectors.

### 2.1 CONTROL ATTRIBUTES

We provide a list of structural attributes that provide explicit and precise control over the graph generation process. These include **number of nodes & edges**, which define the scale of a graph; **number of local bridges**, where a local bridge is an edge that is not part of a triangle in the subgraph, they transfer information between different parts of graphs; **graph density**, which is fraction of edges in the graph, computed as $\frac{e}{v(v-1)}$, where $e$ is the number of edges and $v$ is the number of nodes in the graph; **edge connectivity**, which is the minimum number of edges that must be removed to disconnect the given graph; **node connectivity**, which is the minimum number of nodes that must be removed to disconnect the given graph; **number of maximum cliques**, which is the count of maximal complete subgraphs in a graph; **graph diameter**, which is the length of the shortest path between the

---

[1]The code, data and its splits will be released

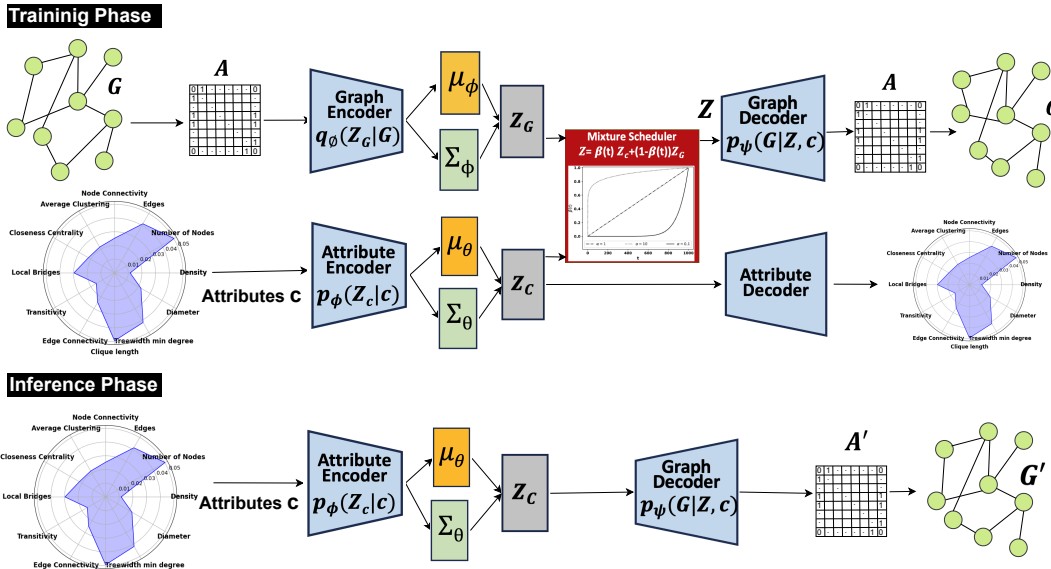

Figure 1: CGRAPHGEN uses both graph attributes and adjacency matrix *during* training for improved decoder tuning. It implements a novel scheduling technique to effectively integrate attributes and graph distributions to provide fine-grained topological control in generation. At test and inference times, it only relies on desired attributes to generate graphs.

most distanced nodes in a graph; **treewidth min degree**, which is an integer quantifying the degree to which a graph deviates from a tree; **closeness centrality**, which is the average distance of a node to all other nodes in the graph, in case of disconnected graphs, to all other nodes in the connected component containing the node, we compute the average of closeness centrality scores across all nodes; **clustering coefficient**, which is the fraction of triangles that exist in a node's neighborhood, we compute the average of clustering coefficients across nodes; and **transitivity**, which is the fraction of all possible triangles present in a graph, computed as $3 \times |triangles|/|triads|$, where a "triad" is a set of three vertices connected by only two edges.

**Importance**    These attributes enable precise control over graph generation and make it possible to generate graphs that satisfy diverse and complex structural requirements. For example, attributes like transitivity and graph density can be adjusted to manage connectivity of graphs, e.g. optimizing the design of local area networks where higher density provides robust communication, or controlling the arrangement of atoms in molecular structures. By controlling these attributes at the granular level, we can generate graphs that satisfy specific needs. This has various applications, generating balanced graph datasets in terms of structural attributes, augmenting small-scale datasets by generating similar but distinct subgraphs; and finding novel structures in specific domains, such as chemistry and molecular biology, where identifying novel compounds with specific properties is crucial for drug discovery.

## 2.2 Learning Representations

**Graph Encoder**    CGRAPHGEN encoder uses a convolution neural networks (CNNs)[2] to encode structural information of graph $G$ into a latent representation $\mathbf{Z_G}$ and obtain parameters for the posterior distribution $q_\phi$. We define this distribution as follows:

$$q_\phi\left(\mathbf{Z_G}|G\right) \quad = \quad \mathcal{N}\left(\mathbf{Z_G}|\mu_\phi\left(G\right), \Sigma_\phi\left(G\right)\right), \tag{1}$$

where $\mathcal{N}$ denotes the Gaussian distribution with parameters $\mu_\phi\left(G\right)$ and $\Sigma_\phi\left(G\right)$ as the mean vector and covariance matrix, obtained from a neural network with parameters $\phi$.

**Attribute Encoder**    To control graph generation, the attribute encoder learns the representation of the attribute vector **c**, such that attribute vector that are similar to each other remains similar in the latent space. We learn the parameters for the prior distribution $p_\theta$ using a normal distribution with the

---

[2]The framework is compatible with GNNs but we focus on CNNs due to better performance in experiments.

mean obtained from a non-linear transformation of $\mathbf{c}$:

$$p_\theta \left( \mathbf{Z_c} | \mathbf{c} \right) \quad = \quad \mathcal{N} \left( \mathbf{Z_c} | \mu_\theta = f \left( \mathbf{c} \right), \Sigma_\theta = I \right), \tag{2}$$

where $f(\mathbf{c})$ is a non-linear transformation of the attribute vector obtained by training a feed forward neural network, and $\Sigma_\theta$ is the unit variance.

## 2.3 MIXTURE-SCHEDULER

Compare to conventional approach of sampling from distributions to bring prior and posterior distributions closer, e.g. through Wasserstein distance (Kantorovich, 1960) or KL divergence (kul, 1951), we introduce MIXTURE-SCHEDULER to integrate the prior $p_\theta$ and posterior $q_\phi$ more effectively–learn better representations that satisfy desired attribute $\mathbf{c}$. Specifically, we define the mixture by combining approximate sample and prior sample as follows:

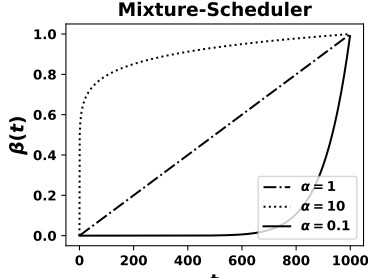

$$\mathbf{Z} = \beta(t)\mathbf{Z_c} + (1 - \beta(t)) \mathbf{Z_G}, \tag{3}$$

where $\beta(t)$ is the inclusion factor at epoch $t$, which controls how much of the prior is incorporated at each stage of training. It is obtained using the following scheduler function:

$$\beta(t) = \min \left( \gamma, (1 - (1 - \beta_0)(1 - t))^{\frac{1}{\alpha}} \right), \tag{4}$$

Figure 2: $\alpha > 0$ controls the inclusion factor while creating final representation $\mathbf{Z}$ from $p_\theta$ and $q_\phi$. It specifies how quickly the prior is integrated during training. A smaller $\alpha$ means less inclusion of $p_\theta$ during the initial epochs and more inclusion toward the end of training.

where $\gamma$ is a value between $[0, 1]$ and sets the maximum possible inclusion from prior $p_\theta$; $\alpha > 0$ specifies how quickly the prior is integrated during training; $t$ represents the current epoch; and $\beta_0$ represents initial inclusion value. The intuition behind developing (4) is to provide flexible control over the contributions of the prior and posterior and allow smooth and gradual transition between them. This approach is principled and helps balance structural and attribute information effectively.

## 2.4 GRAPH GENERATION

We use a Bernoulli distribution to model edge probabilities between node pairs and generate the adjacency matrix $\mathbf{A}$. The graph decoder learn the likelihood distribution $p_\psi$ from $\mathbf{Z}$ to maximize the probability of generating graphs satisfy $\mathbf{c}$:

$$p_\psi \left( G | \mathbf{Z}, \mathbf{c} \right) \sim \text{Bernoulli} \left( D \left( \mathbf{Z} \right) \right), \tag{5}$$

where $D$ is the decoder and 1 from the Bernoulli distribution indicates an edge between a pair of nodes.

**Training Objective** We use the following objective function to learn model parameters:

$$\mathcal{L} \left( \phi, \theta, \psi | G, \mathbf{c} \right) = \mathbb{E}_{q_\phi(\mathbf{Z}|G)} \left[ \log p_\psi \left( G | \mathbf{Z}, \mathbf{c} \right) \right] - \lambda_{WD} \cdot D_{WD} \left( q_\phi \left( \mathbf{Z_G} | G \right), p_\theta \left( \mathbf{Z_c} | \mathbf{c} \right) \right) \tag{6}$$

$$+ \lambda_c \cdot \mathbb{E}_{p_\theta(Z_c|c)} \left[ \left( c - d \left( Z_c \right) \right)^2 \right],$$

where the expected log-likelihood term is the reconstruction loss, which encourages generating graphs that are closer to the given graph $G$ conditioned on $\mathbf{Z}$ and $\mathbf{c}$, the Wasserstein Distance term ($D_{WD}$) regularizes the difference between the approximate posterior $q_\phi(\mathbf{z}|G)$ and the prior $p_\theta(\mathbf{z}|\mathbf{c})$, and $\lambda_{WD}$ and $\lambda_c$ are hyperparameters to balance the regularization terms. The objective encourages reconstruction of attribute vectors $\mathbf{c}$.

**Inference Process** During inference, the model generates a graph conditioned on the desired attribute vector $\mathbf{c}$ using $p_\theta$, see Figure 1, where $p_\theta$ creates a latent representation to set the parameters for the $p_\psi$ distribution to sample and generate graphs.

## 3 EXPERIMENTS

**Datasets**    We use a wide range of datasets for experiments:

*WordNet* (Miller, 1995): a large lexical dataset of English words where nouns, verbs, adjectives and adverbs are grouped into sets of synonyms, and each word represents a distinct concept. Words are connected to each other by different relationships. We considered hypernyms, hyponyms, meronyms, and holonyms to create different WordNet graphs.

*Ogbn-arxiv* (Hu et al., 2020): The Open Graph Benchmark dataset includes a citation network between arxiv papers in computer science, where each node is a paper and an edge represents a citation from one paper to another. In addition, each paper contains an embedding vector obtained from the average of the words present in the title and abstract of the paper.

Table 1: Data statistics in terms of number of graphs and attribute vectors.

|  | **Train** | **Val** | **Test** |
|---|---|---|---|
| **WordNet** | 52,675 | 2,926 | 2,927 |
| **Citeseer** | 1,406 | 78 | 79 |
| **Arxiv** | 47,538 | 2,641 | 2,641 |
| **MUTAG** | 169 | 10 | 9 |
| **MOLBACE** | 1,323 | 74 | 74 |

*Citeseer* (Kipf & Welling, 2017): a citation network of scientific articles, where nodes are papers and edges indicate citations between them.

*MUTAG* (Morris et al., 2020): a molecular dataset where each graph represents a chemical compound and classified as if the given molecule have mutagenic effect on specific gram negative bacterium.

*MOLBACE* (Hu et al., 2020): a molecular dataset where each graph represents a chemical compound.

We create several datasets of graphs by considering the $k$-hop neighbors, $k = \{2, 3\}$ of each node in the above graphs to develop training, validation and test data splits for controlled graph generation. Table1 shows the statistics of the resulting datasets.

**Settings**    Following previous works (De Cao & Kipf, 2018; Zahirnia et al., 2024), we set the maximum number of nodes to $V = 50$ in experiments. This threshold is appropriate for GNNs due to the nature of how GNNs process graph data, especially when considering the common practice of sampling 1-2 hop neighbors form localized subgraphs for nodes. We set the number of hops to $k = 2$ for all datasets except for Citeseer, for which we use $k = 3$ due to its smaller size. In addition, we use the Networkx package (Hagberg et al., 2008) to obtain graph attributes. We consider maximum number of 1000 training iteration for Citeseer and 200 iterations for other datasets. We run all experiments on a single A100 40GB GPU. Other settings are detailed in Appendix 6.1.

**Evaluation Metrics**    We use mean absolute difference (MAD↓) metric for evaluation. MAD computes the absolute difference between the attributes of predicted graphs and their corresponding target graphs. We average these differences for each dataset.

**Baselines**    We compare CGRAPHGEN against the following baselines. For fair evaluation, we incorporate our control attributes to all baseline models except GraphRNN which is a free generative model. We provide it's performance for reference.

*GraphRNN* (You et al., 2018): GraphRNN generates graph iteratively by training on a representative set of graphs using breath first search of nodes and edges and implements node and edge RNNs to generate target graphs. GraphRNN is not a controlled generation approach.

*EDGE* (Chen et al., 2023): is a diffusion based generative model which iteratively removes edges to create a completely disconnected graph and uses decoder to iteratively reconstruct the original graph. It explicitly uses adjacency matrix to satisfy the statistics of the generated graphs during training.

*GenStat* (Zahirnia et al., 2024): learns the latent adjacency matrix conditioned on graph level attributes, and decodes it to recreate attribute statistics and use them to generate graphs.

### 3.1 MAIN RESULTS

Table 2 shows the overall performance of models across datasets. CGRAPHGEN achieves a lower MAD (↓) compared to other baselines, which indicates that its generated graphs more accurately satisfy the specified topological properties.

Table 2: Performance of CGRAPHGEN compared with baselines models. Average mean absolute difference, MAD(↓), is the average of absolute mean error in satisfying target attributes.

| | WordNet | Citeseer | Ogbn-Arxiv | MUTAG | MOLBACE | Average |
|---|---|---|---|---|---|---|
| | MAD | MAD | MAD | MAD | MAD | MAD |
| **GraphRNN** (You et al., 2018) | 3.26 | 5.05 | 4.80 | 1.71 | 3.81 | 3.73 |
| **GenStat** (Zahirnia et al., 2024) | 4.11 | 5.34 | 5.53 | 4.14 | 3.05 | 5.20 |
| **EDGE** (Chen et al., 2023) | 3.91 | 4.97 | 5.52 | 2.62 | 3.07 | 4.16 |
| **CGRAPHGEN** | **1.80** | **1.71** | **2.14** | **1.00** | **1.90** | **1.71** |

EDGE outperforms GenStat in controlled graph generation. This is mainly because EDGE explicitly models adjacency matrix, whereas GenStat treats adjacency matrix as a latent variable. In contrast, CGRAPHGEN generates graphs from a structure-aware distribution conditioned on attributes (and on graphs during training). Thus, during inference process, CGRAPHGEN is able to generate graphs with attributes that closely match the specified ones, and result in lower MAD scores.

**Output Visualization** Table 3 shows examples of different graphs generated by CGRAPHGEN, GenStat, and EDGE across datasets. The value under each graph indicates the MAD score between generated and test graphs. As evident from the Table, CGRAPHGEN generates graph that are more similar to the target graphs compared to other baseline models. We attribute this improvement to CGRAPHGEN's ability to perform fine-grained controlled generation using graph attributes.

Table 3: Graph visualization across datasets. Examples are taken from test splits of datasets.

| | Wordnet | | Citeseer | | Ogbn-Arxiv | | Mutag | | Molbace | |
|---|---|---|---|---|---|---|---|---|---|---|
| **CGRAPHGEN** MAD | 0.85 | 0.00 | 0.10 | 2.36 | 2.85 | 3.71 | 0.76 | 1.09 | 0.51 | 0.68 |
| **GenStat** MAD | 2.61 | 8.44 | 2.67 | 7.11 | 9.32 | 1.28 | 4.01 | 1.67 | 6.09 | 0.84 |
| **EDGE** MAD | 3.95 | 0.56 | 15.32 | 10.6 | 7.29 | 3.52 | 0.88 | 3.01 | 4.82 | 2.44 |

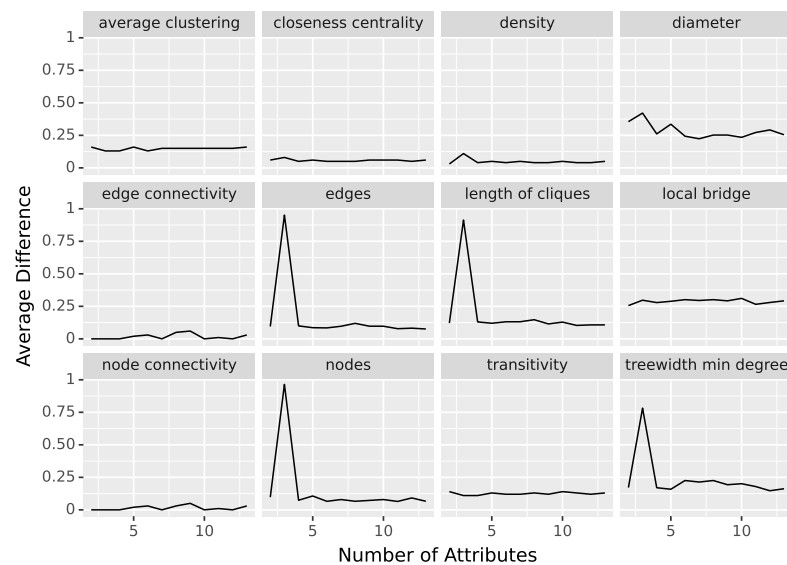

Figure 3: The "average difference" error trend for each attribute as additional attributes are gradually included in each subsequent independent training run till all the attributes are covered. The x-axis shows the number of attributes considered for each run. Average difference is the normalized error between gold standard attributes and attributes calculated from generated graph. As the number of attributes increases, the error decreases and gradually stabilizes for each attribute. This shows that fine-grained attributes, beyond just the number of nodes and edges, are crucial for generating graphs that satisfy the specified attributes. As more constraints are added in using new attributes, the structural quality of the generated graph improves.

## 4 DISCUSSION

We conducted several ablation studies to understand the effectiveness of CGRAPHGEN in controlled graph generation. We analyze scalability to larger number of nodes; provide insights on generating graphs by masking fundamental attributes like number of nodes and edges, while providing all other fine-grained attributes; and provide a detailed study on MIXTURE-SCHEDULER, where we analyze the effects of limiting the inclusion factor and varying the rate of inclusion.

**Scalability to larger graphs**    We analyse the effect of increasing the maximum number of nodes, $|V|$, on CGRAPHGEN's MAD performance. Table 4 shows that MAD increases as the maximum number of nodes grows. This is because larger graphs have greater structural complexity, with more potential edges and relationships that are harder to generate accurately. This makes it challenging for the model to capture both local and global topological properties, and potentially leads to cumulative errors in matching node-specific attributes like degrees and centrality. In addition, larger graphs often contain more variability and sparsity, which further complicates satisfying the desired structural attributes and result in higher deviations between the generated and target graphs.

Table 4: MAD increases as number of nodes grows.

| #Nodes | MAD($\downarrow$) |
|---|---|
| 60 | 5.13 |
| 80 | 12.98 |
| 100 | 25.93 |
| 200 | 31.90 |

**More control attributes improve results**    Graph attributes determines the structural details for generated graphs. Figure 3 shows the effect of gradually adding more attributes during training. Starting with two basic graph attributes (number of nodes and edges), we retrain the model while adding one randomly selected attribute at a time. and report the trend of average difference. As Figure 3 shows, as the number of control attributes increases, the error decreases and stabilizes, which indicates that CGRAPHGEN learns more about graph structure and generates more accurate graphs using more fine-grained attributes. We believe generating graph using only nodes and edges can results in multiple possibilities of the graphs with different structural properties, which gives more freedom

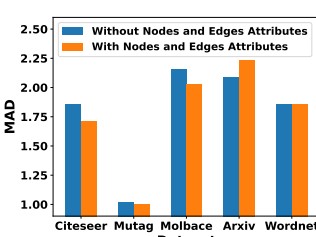 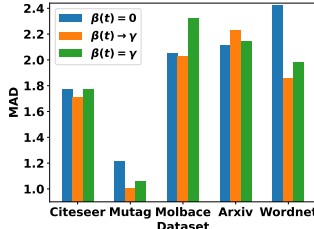 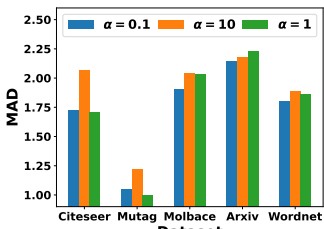

(a) Performance of CGRAPHGEN without using number of nodes & edges as controls.

(b) Effect of inclusion factors on generation error across different datasets.

(c) Effect of $\alpha$ on error in generating graphs across different datasets.

Figure 4: Ablation Analysis

to the model and allows it to differ from desired graph. However, as more restrictive constraints are enforced like density or closeness centrality, the quality of the generated graph improves.

**Generation without number of nodes and edges as attributes**  Figure 4(a) compares the performance of CGRAPHGEN before and after masking two basic graph attributes (number of nodes and edges as controls) for training the model. The model performs almost similarly without these attributes, which indicates that CGRAPHGEN can infer the number of nodes and edges with minimal error based on other fine-grained attributes.

## 4.1 MIXTURE-SCHEDULER ANALYSIS

We conduct ablation study of MIXTURE-SCHEDULER to answer following questions: (Q1) Does including the prior distribution $p_\theta$ help? (Q2) How does the rate of inclusion affect model's performance? (Q3) How much of the prior should be included?

**Q1: Does including the prior distribution $p_\theta$ help?**  To understand the effect of using MIXTURE-SCHEDULER, we consider three scenarios: (i) when the model only learns from $q_\phi$ distribution ($\beta(t)$ = 0), (ii) when the model gradually combine $p_\theta$ and $q_\phi$ as training progresses ($\beta(t) \to \gamma$), and (iii) when the model combines both $p_\theta$ and $q_\phi$ with constant influence factor $\beta(t) = \gamma$. As shown in Figure 4(b), combining representations from both distributions $p_\theta$ and $q_\phi$ helps to generate better graphs compared to using only representations from $q_\phi$. In addition, gradual increase in influence factor $\beta(t) \to \gamma$ performs better compared to keeping it constant $\beta(t)$. We conclude relying only on graph representation from $q_\phi$ without considering attribute representation from $p_\theta$ results in higher MAD error and lower performance.

**Q2: How does the rate of inclusion affect model's performance?**  To answer the second question, we analyze different rates of inclusion. As Figure 4(c) shows, a slow inclusion rate ($\alpha = 0.1$) often helps model in learning better representations compared faster inclusion rates, e.g. ($\alpha$=10). This result suggests that initially focusing on the $q_\phi$ and gradually incorporating the $p_\theta$ yields better latent representations.

**Q3: How much of the prior should be included?**  To understand the effect of combining attributes representation from $p_\theta$ with graph representations from $q_\phi$, we vary the influence of prior distribution using different values of maximum possible inclusion, $\gamma$. We set $\gamma$ from [0,1] with step size of 0.1. When $\gamma$=0, there is no influence from $p_\theta$, and when $\gamma = 1$, no information from posterior $q_\phi$ is considered. Any values in between combines information from $p_\theta$ and $q_\phi$ distributions. Figure 5 shows that smaller values of $\gamma$ result in lower error, which indicates that a limited inclusion of the $p_\theta$ helps generate better graphs by controlling the contributions of both distributions.

## 4.2 DE-NOISING GRAPH ATTRIBUTES

We evaluate CGRAPHGEN's robustness to noisy attributes by masking one attribute at a time during inference. For this experiment, we use our best trained model and freeze its parameters. Then, we run inference 12 times, once per attribute and during each run we set the attribute value to zero while

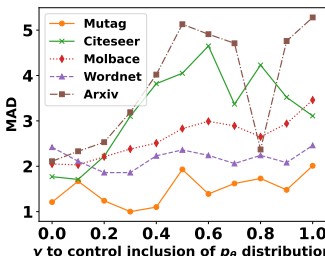
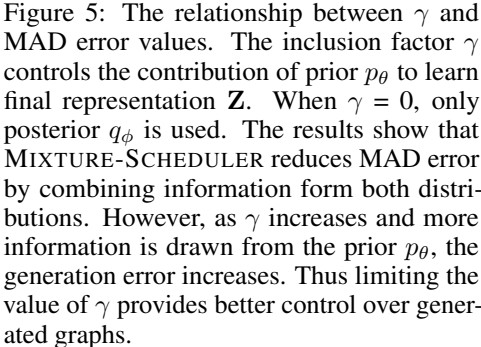
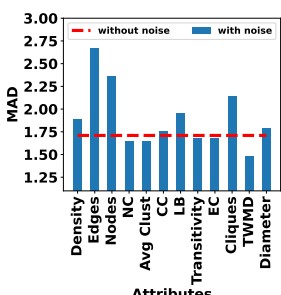

Figure 5: The relationship between $\gamma$ and MAD error values. The inclusion factor $\gamma$ controls the contribution of prior $p_\theta$ to learn final representation $\mathbf{Z}$. When $\gamma = 0$, only posterior $q_\phi$ is used. The results show that MIXTURE-SCHEDULER reduces MAD error by combining information form both distributions. However, as $\gamma$ increases and more information is drawn from the prior $p_\theta$, the generation error increases. Thus limiting the value of $\gamma$ provides better control over generated graphs.

Figure 6: MAD on test data when masking only one attribute with noise while keeping others unchanged. The dotted line shows CGRAPHGEN's performance on Citeseer without any masking. Each bar shows MAD when when a specific attribute is masked. Abbreviations NC (node connectivity), EC (edge connectivity), TWMD (tree width min degree), Avg Clust (average clustering), LB (number of local bridge), Clique (number of cliques).

keeping other attributes unchanged. We run the model on the entire test graphs. Figure 6 shows the results, where the dotted horizontal line shows the MAD error value of CGRAPHGEN without masking any attribute and serves as a reference to compare against each independent inference run. The results show that CGRAPHGEN is often able to ignore noisy attributes and generate accurate graphs based on the remaining attributes, which demonstrates its resilience to missing control attributes.

## 5 CONCLUSION AND FUTURE WORK

We presented CGRAPHGEN, a novel approach to controlled graph generation that generates graphs satisfying fine-grained topological attributes. CGRAPHGEN introduces a novel scheduling technique, MIXTURE-SCHEDULER, which effectively combines *attribute* representations with *adjacency matrix* representations to learn accurate latent representations for graph generation *during training*. It enables precise control over generated graphs, even without explicitly specifying some of the basic graph properties such as the number of nodes and edges. Our experiments demonstrate that CGRAPHGEN produces graphs with lower error by integrating structural information gradually and leveraging multiple control attributes. In future, we aim to extend CGRAPHGEN to handle dynamic or temporal graphs, where the graph structure evolves over time for applications in social network analysis, traffic prediction, and temporal knowledge graphs.

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

## 6 APPENDIX

### 6.1 SETTINGS

We set $\gamma$ to 0.3 for Mutag, 0.1 for Molbace, Citeseer, and arxiv; and 0.2 for Wordnet dataset. For the CNN encoder, we used two layers of CNN with kernel size of 5 and 32, 64 channels respectively. For the decoder, we used two layers of CNN with 64,32 channels respectively. We consider a batch-size of 1,028.

