# OpenReview forum: "A VARIATIONAL FRAMEWORK FOR GRAPH GENERATION WITH FINE-GRAINED TOPOLOGICAL CONTROL"
_ICLR.cc/2025/Conference — Submitted to ICLR 2025_

### Official Review · Reviewer_yNfm · 2024-10-30

**Soundness:** 3
**Presentation:** 3
**Contribution:** 2
**Rating:** 3
**Confidence:** 5

**Summary:**

The paper proposes CGRAPHGEN, a novel conditional variational autoencoder framework for generating graphs with fine-grained control over topological attributes. The framework introduces a MIXTURE-SCHEDULER, a scheduling technique to combine structural and attribute-based latent representations. Experiments on multiple datasets show that CGRAPHGEN outperforms baseline models.

**Strengths:**

- The model offers flexibility in controlling multiple structural properties (e.g., graph density, connectivity, clustering coefficient), enabling accurate graph generation across various domains.
- The mixture-scheduler seems to be novel and it smoothly integrates prior and posterior distributions, improving the quality and stability of generated graphs.

**Weaknesses:**

- The proposed model doesn't seem to be much of an improvement compared to GraphVAE-like models. The condition architecture is very common in generative models, and feature/attribute based conditional graph generation seems to be a common trick in most methods. Therefore, I think the proposed model may lack enough novelty.
- Lack of baselines. I have noticed this paper include the diffusion-based model (EDGE), why not other SOTA graph generative models like DruM, DIGress and so on. For graph generation, I think it is more convincing to compare these models or at least other vae-based models. As far as I know, I believe these models can also incorporate the attribute feature to achieve conditional graph generation.
- For the mixture-scheduler part, I don't really understand the meaning of regarding the time $t$ as epoch in the training stage. From Figure 4(b), it seems there is no clear effect on whatever the $\beta(t)$ is.
- In your ablation, I find the experiments with masked only one attribute, is there any flexibility attributes choice?

[1] Efficient and degree-guided graph generation via discrete diffusion modeling.
[2] Graph generation with diffusion mixture.
[3] Discrete denoising diffusion for graph generation.

**Questions:**

- Is there any code for the proposed model?
- Have you tried other more complex neural network architecture?

---

> ### Author Response · Authors · 2024-11-23
>
> **Q.** The proposed model doesn't seem to be much of an improvement compared to GraphVAE-like models. The condition architecture is very common in generative models, and feature/attribute based conditional graph generation seems to be a common trick in most methods. Therefore, I think the proposed model may lack enough novelty.
>
> **A.** The novelty of our approach is in introducing a novel scheduling technique to effectively integrate representations from adjacency matrices and attribute distributions in a conditional variational autoencoder, which enables fine-grained and precise control in graph generation. These ideas have not been investigated before.
>
> **Q.** Lack of baselines. I have noticed this paper includes the diffusion-based model (EDGE), why not other SOTA graph generative models like DruM, DIGress and so on. For graph generation, I think it is more convincing to compare these models or at least other vae-based models. As far as I know, I believe these models can also incorporate the attribute feature to achieve conditional graph generation.
>
> **A.** There is limited work on controlled graph generation—models that generate graphs that optimally satisfy desired topological attributes provided as input. The references suggested by the reviewer are not designed to steer graph generation based on “desired” topological properties. Rather, they generate graphs either unconditionally  or conditioned on class labels and cannot take desired structural properties as input. Our baselines models use graph topological properties to generate graphs and we found those to be the closest possible baselines for controlled graph generation, where desired structural properties are given to the generation model as input. Specifically, GENSTAT (2023) generates graphs conditioned on the given values of the desired structural properties, EDGE (2023) is the diffusion-based generation model that generates graphs based on given distributions properties.
>
> **Q.** For the mixture-scheduler part, I don't really understand the meaning of regarding the time t as epoch in the training stage. From Figure 4(b), it seems there is no clear effect on whatever the β(t)  is. In your ablation, I find the experiments with masked only one attribute, is there any flexibility attributes choice?
>
> **A.** Figure 4(b) shows the effect of inclusion factors β(t)  on generation error across different datasets. It is indicated in the Figure that when β(t) = 0 i.e. no inclusion, the error is high compared to when β(t) -> $\gamma$ (when inclusion increases as time/epochs increases) and when β(t)  = $\gamma$ (with constant inclusion factor with respect to time/epochs). Also, when  β(t) -> $\gamma$, the error rate is lower compared to when β(t)  = $\gamma$.
> In addition, any  desired attributes can be selected based on the desired topological  structure. The attributes we investigated are not exhaustive, but the model is applicable to any number of attributes.
>
> **Q.** Have you tried other more complex neural network architecture?
>
> **A.** Yes, we tried several GNN models (GraphSAGE, GTN, and GIN) but the current implementation works best.
>
> **Q.** Is there any code for the proposed model?
>
> **A.** Certainly. As a common practice, we will release our code and data splits before releasing the camera-ready of each paper.

---

> > ### Comment · Reviewer_yNfm · 2024-11-25
> >
> > Thanks for your response! I still have the issue about isn't it possible to encode these topological features or use them directly on more baselines as conditions?

---

> ### Author Response · Authors · 2024-11-26
>
> It is indeed possible to use topological features as conditions for baselines, and we have done so by incorporating our control attributes into EDGE [1] and GenStat [2] baselines. However, these heuristic modifications do not address the fundamental limitations of these methods. Existing baselines (including SPECTRE [3] ) lack a principled mechanism, like our scheduling approach, to dynamically integrate and balance attribute-based control during the generation process. Our scheduling technique systematically balances attribute-driven and structure-driven components throughout the training process. This enables precise and robust graph generation, as demonstrated in our experiments. Encoding topological features into the baselines can indeed improve their performance, but it does not match the fine-grained control and adaptability provided by our model.  Again, EDGE,  GenStat and SPECTRE are the only available baselines for controlled graph generation. We did not include SPECTRE because SPECTRE is computationally expensive to run. In fact, compared to the datasets used in SPECTRE, our datasets contain more graphs with a much larger number of nodes. Therefore, we didn’t include SPECTRE in experiments.
>
>  [1] Efficient and degree-guided graph generation via discrete diffusion modeling. \
> Xiaohui Chen, Jiaxing He, Xu Han, and Liping Liu. \
> In International Conference on Machine Learning, PMLR, 2023. \
> URL https://proceedings.mlr.press/v202/chen23k/chen23k.pdf.
>
>  [2] Neural graph generation from graph statistics. \
> Kiarash Zahirnia, Yaochen Hu, Mark Coates, and Oliver Schulte. \
> Advances in Neural Information Processing Systems, 2023.\
> URL https://proceedings.neurips.cc/paper_files/paper/2023/file/72153267883fbcafdb6e4662382696c5-Paper-Conference.pdf.
>
>  [3] Spectre: Spectral conditioning helps to overcome the expressivity limits of one-shot graph generators. \
> Martinkus, Karolis and Loukas, Andreas and Perraudin, Nathanael and Wattenhofer, Roger.\
> International Conference on Machine Learning, PMLR, 2022.\
> URL https://proceedings.mlr.press/v162/martinkus22a.html

---

### Official Review · Reviewer_Df1K · 2024-11-03

**Soundness:** 3
**Presentation:** 3
**Contribution:** 3
**Rating:** 6
**Confidence:** 2

**Summary:**

This paper proposes a conditional variational autoencoder for graph generation with fine-grained topological control. The proposed model incorporates a scheduling technique to integrate representations from both the adjacency matrix and attribute distribution to enable fine-grained control.

**Strengths:**

1. This paper proposes a new setting for the controlled graph generation task, which is highlighted by the injection of fine-grained topological control.
2. The proposed method seems technically sound to me.

**Weaknesses:**

1. The number of baseline models compared in the experiments appears to be limited.
2. I'm not sure if it's reasonable to use only the MAD metric to evaluate the generation results based on various topological attributes.

**Questions:**

Please refer to the Weaknesses.

---

> ### Author Response · Authors · 2024-11-23
>
> **Q.** The number of baseline models compared in the experiments appears to be limited.
>
> **A.** There is limited work on controlled graph generation—models that generate graphs that optimally satisfy desired topological attributes provided as input. In addition to our baselines GENSTAT (2023) and EDGE (2023), there is SPECTRE (2022), which is a diffusion-based method for controlled graph generation. We didn’t include this model because SPECTRE is computationally expensive to run. In fact, compared to the datasets used in SPECTRE, our datasets contain more graphs with a much larger number of nodes. Therefore, we didn’t include SPECTRE in experiments.
>
> **Q.** I'm not sure if it's reasonable to use only the MAD metric to evaluate the generation results based on various topological attributes.
>
> **A.** The main goal of the controlled generation is to satisfy given topological attributes. MAD computes the absolute difference between the attributes of predicted graphs and their corresponding target graphs. We believe this metric quantifies the quality of generation, as we visualized in Table 3. Other metrics such as graph edit distance are expensive for larger graphs (beyond 10 nodes).

---

> > ### Comment · Reviewer_Df1K · 2024-12-03
> >
> > Thank you for your response. I appreciate the discussion you provided regarding my concerns. After reviewing your explanations, I have decided to maintain my score.

---

> > > ### Author Response · Authors · 2024-12-04
> > >
> > > Thank you for taking the time to read our response. We appreciate any feedback and would love to understand better—could you kindly share which parts of our response didn’t fully address your concerns? This would help us improve further. Thank you!

---

### Official Review · Reviewer_XWvC · 2024-11-04

**Soundness:** 1
**Presentation:** 2
**Contribution:** 2
**Rating:** 3
**Confidence:** 4

**Summary:**

The paper focuses on controlled graph generation that generates graphs satisfying specific topological attributes. It introduces a new scheduling technique, MIXTURE-SCHEDULER, to combines desired attributes with adjacency matrix representations during training for precise graph generation, and it then uses only attributes during inference. Experiments demonstrate that generated graphs have better aligned attributes.

**Strengths:**

1. mixing the attributes and graph representation in latent space for VAE is somehow new for controlled generation.
2.  the results for controlled graph generation is seemingly good regarding attribute alignment.

**Weaknesses:**

1. The biggest concern is the paper lacks of a rigorous deduction for the VAE model and learning objective. For most VAEs, we generally start from the miminization of log likelihood and use variational inference to factorize it. However, the formulations in this paper are very heuristic. We do not know whether the mixing of attributes and graph representation is valid. Mixing the prior with posterior looks also weird to me. What I expect should be starting something like $P(G|c) = \int_{Z_G, Z_c} P(G|Z_G, Z_c, c)P(Z_G|\theta, c) P(Z_c|c) dZ_G dZ_c$.
2. It seems the graph encoder/decoder can only deal with adjacency matrix, but how about graphs with node features?
3. The evaluation only measures the attributes, but the validness of the graph in many domains is also important (e.g. for molecules).

**Questions:**

What is d(Z_c) in Eq. (6)? There is no explanation for this notation.

---

> ### Author Response · Authors · 2024-11-23
>
> **Q.** The biggest concern is the paper lacks of a rigorous deduction for the VAE model and learning objective. For most VAEs, we generally start from the minimization of log likelihood and use variational inference to factorize it. However, the formulations in this paper are very heuristic. We do not know whether the mixing of attributes and graph representation is valid. Mixing the prior with posterior looks also weird to me. What I expect should be starting something like
> P(G|c)=∫ZG,ZcP(G|ZG,Zc,c)P(ZG|θ,c)P(Zc|c)dZGdZc
>
> **A.** We thank the reviewer for raising this question. We would like to clarify our approach and provide the rationale behind it. We start with modeling the conditional distribution P(G|c), where c represents the attributes. In our formulation, we decompose the latent variable into two components: one representing structural information obtained from the adjacency matrix (only during training) and another representing attribute information. We approximate the posterior distribution of the latent variables with a variational distribution, which balances the reconstruction of the graph and attributes and aligns the prior (attribute-based representation) and posterior (structural representation) distributions by generating graphs that satisfy the specified attributes. The mixture representation is carefully designed to minimize the gap between the prior and the posterior. The proposed scheduling technique allows for a gradual and principled transition during training, essentially, moving from a posterior-dominated representation to a prior-dominated latent space. This way of mixing prior and posterior representations may appear unconventional, but it indeed aligns graph structure and attributes effectively in the latent space, please see Figure 5 ($\gamma$=0.0).
>
> **Q.** It seems the graph encoder/decoder can only deal with adjacency matrix, but how about graphs with node features?
>
> **A.** Node features can be handled by the graph encoder as in a standard GNN without any modification to the proposed architecture. Our architecture is not limited to a specific type of encoder/decoder models. We will add this information to the paper.
>
> **Q.** The evaluation only measures the attributes, but the validness of the graph in many domains is also important (e.g. for molecules).
>
> **A.** The goal of controlled graph generation is to create graphs that optimally satisfy the desired topological attributes provided as input. It not only aims to generate valid graphs within a domain—such as molecules adhering to chemical rules—but also generate "invalid" graphs, such as those that violate these rules. The ability to generate such "invalid" graphs can provide insights, particularly in domains like biology, where understanding non-conforming structures can be important. In addition, defining a universally "valid" graph is impractical or impossible in many contexts. Therefore, our focus is on satisfying the specified topological constraints and attributes during the generation process.
>
>
> **Q.** What is d(Z_c) in Eq. (6)? There is no explanation for this notation.
>
> **A.** d(Z_c) in Eq. (6) represents the decoder of Z_c. We will add this information in the paper.

---

### Official Review · Reviewer_vgXE · 2024-11-04

**Soundness:** 2
**Presentation:** 2
**Contribution:** 2
**Rating:** 5
**Confidence:** 5

**Summary:**

The paper introduces CGRAPHGEN, a novel framework for controlled graph generation that allows for fine-grained control over graph topological properties. The authors propose a conditional variational autoencoder (VAE) that, unlike previous approaches, utilizes both the graph adjacency matrix and attribute vectors during training for improved decoder tuning and relies only on attributes during inference. This enables CGRAPHGEN to generate graphs that closely match the specified structural attributes.

**Strengths:**

1. The method utilizes a conditional VAE that integrates information from both the adjacency matrix and attribute vectors during training, resulting in more precise graph generation.

2. The scalability of the method is quite good. The method can be used to generate large-scale graphs, which is quite competitive compared to other auto-regression models.

3. The paper is well-written and easy to understand.

**Weaknesses:**

1. The baselines and the datasets are quite simple. The authors are recommended to compare with more recent graph conditional generation methods. e.g. [1] [2] [3]

[1] Yang, Carl, et al. "Conditional structure generation through graph variational generative adversarial nets." Advances in neural information processing systems 32 (2019).
[2] Ommi, Yassaman, et al. "Ccgg: A deep autoregressive model for class-conditional graph generation." Companion Proceedings of the Web Conference 2022. 2022.
[3] Mo, Zhanfeng, Tianze Luo, and Sinno Jialin Pan. "Graph principal flow network for conditional graph generation." Proceedings of the ACM on Web Conference 2024. 2024.

2. it is unclear how the hyper-parameters are defined. In Figure 5, the performance seems quite stable for different gamma, e.g. there's a drop when gamma = 0.8 on arxiv dataset. "When γ increases and more information is drawn from the prior pθ, the generation error increases." is not always true.

3. No theoretical analysis of how the proposed method can reduce the generation error better than other baseline methods.

**Questions:**

Please refer to the Weaknesses.

---

> ### Author Response · Authors · 2024-11-23
>
> **Q.** The baselines and the datasets are quite simple. The authors are recommended to compare with more recent graph conditional generation methods. e.g. [1] [2] [3]
>
> **A.** There is limited work on controlled graph generation—models that generate graphs that optimally satisfy desired topological attributes provided as input. All the three references suggested by the reviewer are not designed to steer graph generation based on “desired” topological properties. Rather, they generate graphs conditioned on class labels and cannot take desired structural properties as input. Our baselines models use graph topological properties to generate graphs and we found those to be the closest possible baselines for controlled graph generation, where desired structural properties are given to the generation model as input. Specifically, GENSTAT (2023) generates graphs conditioned on the given values of the desired structural properties, EDGE (2023) is the diffusion-based generation model that generates graphs based on given distributions properties.
>
> **Q.** it is unclear how the hyper-parameters are defined. In Figure 5, the performance seems quite stable for different gamma, e.g. there's a drop when gamma = 0.8 on arxiv dataset. "When $\gamma$ increases and more information is drawn from the prior pθ, the generation error increases." is not always true.
>
> **A.** Figure 5 shows our analysis on the best inclusion value ($\gamma$) using MIXTURE-SCHEDULER. It shows significant differences in MAD values (between 1 to more than 5) for different inclusion rates. In addition, the results show that the MAD trend for all the datasets tend to increase when more information is drawn from the prior pθ. The exact values of $\gamma$ and their corresponding MAD scores are as follows:
>
>
> | $\gamma$|Mutag (MAD)|Citeseer (MAD) |Molbace (MAD) |Wordnet (MAD) |Arxiv(MAD) |
> |--------|:--------:|:--------:|:--------:|:--------:|:--------:|
> |   0.0  |  1.21  |  1.77  |  2.05  |  2.42  |  2.11  |
> |   0.1  |  1.67  |  1.71  |  2.03  |  2.11  |  2.33  |
> |   0.2  |  1.24  |  2.22  |  2.21  |  1.86  |  2.53  |
> |   0.3  |  1.00  |  3.10  |  2.38  |  1.86  |  3.19  |
> |   0.4  |  1.10  |  3.82  |  2.51  |  2.23  |  4.02  |
> |   0.5  |  1.93  |  4.05  |  2.83  |  2.36  |  5.13  |
> |   0.6  |  1.39  |  4.65  |  2.99  |  2.24  |  4.91  |
> |   0.7  |  1.62  |  3.37  |  2.89  |  2.06  |  4.71  |
> |   0.8  |  1.73  |  4.23  |  2.65  |  2.24  |  2.37  |
> |   0.9  |  1.48  |  3.52  |  2.94  |  2.08  |  4.76  |
> |   1.0  |  2.01  |  3.11  |  3.46  |  2.46  |  5.28  |
>
>
>
> The above results show increasing trends of MAD error as the value of $\gamma$ increases. There are some values of $\gamma$, for example $\gamma$ = 0.8 for arxiv, where the MAD error decreases compared to its previous few $\gamma$ values but it is still higher than the lowest MAD error value found before that $\gamma$.
>
> **Q.** No theoretical analysis of how the proposed method can reduce the generation error better than other baseline methods.
>
> **A.** We provide theoretical intuition to support why our approach is expected to reduce generation error compared to baseline methods. Our model introduces a novel scheduling technique that effectively integrates structural (posterior) and attribute (prior) representations so that graphs with similar topological attributes and structures are mapped closer together in the latent space. The scheduling technique allows for a gradual and principled transition during training, essentially, moving from a posterior-dominated representation to a prior-dominated latent space, which allows our model to learn a smoother and more consistent latent space and reduce the mismatch between the generated graphs and the desired attributes. None of the existing baselines incorporate a dynamic mechanism to balance attribute-driven and structure-driven generation.

---

### Meta-Review · Area_Chair_DBS2 · 2024-12-18

**Metareview:**

In this submission, the authors essentially proposed a conditional VAE framework for graph generation. However, the reviewers have concerns about the organization and writing of this work, which does not explain the deduction of the VAE model clearly, and thus, makes the rationality of the proposed method questionable. In addition, the datasets like MUTAG and MOLBACE are over-simplified. The feasibility of the proposed method on large-scale attributed molecular graph datasets has not been verified. The rebuttals of the authors did not resolve the concerns completely. All the reviewers require the authors to polish the paper and enhance the experimental part.

**Additional Comments On Reviewer Discussion:**

Two of the four reviewers interacted with the authors and, finally, decided to maintain their scores. In the discussion phase, AC asked for additional comments but did not get any feedback till Dec. 18. After reading the submission, the comments, and the rebuttals, AC has decided to reject this work.

---

### Decision · Program_Chairs · 2025-01-22

Reject